# The Harmonization between Religious Freedom and the Protection of Public Health: Betwixt Self-Regulation and Law

**Ioannis E. Kastanas**

Department of Law, School of Law, University of Nicosia, Nicosia 2417, Cyprus; kastanas.i@unic.ac.cy

**Abstract:** The enshrinement of religious freedom in a State Constitution is determined by the system of relations between the State and Religions. A particular aspect of religious freedom is freedom of worship, which was reframed due to the COVID-19 pandemic with the adoption of measures for the protection of public health. The examples of Cyprus and Greece demonstrate that the self-regulation enjoyed by religious communities may be harmonized with the need to protect public health and is directly connected to the existing system of relations between the State and Religions. The case law on the State restrictive measures regarding worship in conjunction with the measures implemented by the religious communities themselves together give rise to the general principles of harmonization between religious freedom and the protection of public health, with respect for the principle of proportionality.

**Keywords:** religious freedom; COVID-19; autonomy; religious communities

## 1. Introduction

Carl Schmitt begins his work Political Theology with a claim: "Sovereign is he who decides on the exception" (Schmitt 1985, p. 5). A state of emergency, being an exceptional, extraordinary situation, which can occur in the life of every state, poses numerous challenges to the government. As this way of state functioning necessarily leads to the temporary concentration of power aimed at overcoming a crisis, there is always a fear of its resulting in the suspension of some of the constitutional provisions to an extent greater than necessary. The harmonization between the individual freedoms and the common good is one of the crucial stakes in the external periods. The type of harmonization differentiates the democratic societies from the autocratic regimes. The pandemic caused by the spread of the COVID-19 virus necessitated the adoption by the vast majority of states worldwide of harsh restrictive measures to halt it. Such measures were imposed on the exercise of religious worship (Dieu 2021, pp. 173–80) among other practices. The imposition of, occasionally notably harsh, restrictions on worship rekindled the discussion[1] on the relation between religious freedom and the protection of public health. The present paper aims to identify the principles of their harmonization, which are directly connected to the relations between the State and Religions in each state. As such, two salient examples were selected for discussion: Cyprus and Greece. They are two separate States, whose Orthodox-majority peoples share a spiritual bond. However, while the local Churches (Church of Cyprus and Church of Greece) belong to the same confession and constitute parts of the administrative structure of the Eastern Orthodox Church[2], they are nevertheless self-administered. Furthermore, the history of both Churches is intertwined with the history of the State in which they exist and exerts influence on the nature of the relationship between Church and State, encompassing how religious freedom is protected. It is therefore reasonable that any restrictions on religious freedom imposed over the course of the pandemic and the harmonization of that fundamental right with public health should pass through the system of Church–State relations. This paper will utilize productive as well as deductive reasoning.

To begin, their regulatory frameworks governing the relations between the State and Religions will be discussed; subsequently, the measures their Governments implemented and their review by the Courts as well as the measures implemented by religious communities themselves will be analyzed. We will thus be able to identify by induction the general principles of the harmonization between religious freedom and the protection of public health.

**2. The Safeguarding of Religious Freedom within the Context of the Relations between the State and Religions in Cyprus and Greece**

Freedom of religion is enshrined in the constitutions of Cyprus and Greece alike. However, this enshrinement is influenced by the existing system of relations between the State and Religions in each of these countries. Indeed, the extent of protection at the national level does not present substantial variation from its enshrinement in the ECHR and the Charter of Fundamental Rights of the European Union.

*2.1. Cyprus*

According to a number of provisions of the Constitution of the Republic of Cyprus[3] (henceforth CC), no religion is enshrined as official or prevailing[4] (Emilianides 2019, p. 95). The CC accords privileged status to the five major religions of the island, i.e., the Orthodox Church (to which the majority of the population belongs), the Muslim faith, the Maronite Church and the Armenian and Roman Catholic Churches, albeit without making distinctions or classifications in the level of protection. Furthermore, the State is not religious: State officials do not swear a religious oath when assuming their duties, instead solemnly affirming their *"faith to, and respect for, the Constitution and the laws made thereunder, the preservation of the independence and the territorial integrity, of the Republic of Cyprus, in accordance with articles 42(1), 59(4) and 100 of the Constitution (respectively for the President and the Vice-President of the Republic, the ministers, the members of the House of Representatives and the members of the Communal Chambers)"* (Konidaris and Emilianides 2016, p. 220).

It follows from the above that the CC established a system of homotaxy between the Republic and the five major religions of the island,[5] a hallmark of which is equality between the State and the major religions. As A.C. Emilianides has observed (Konidaris and Emilianides 2016, p. 221) "the five major religions of Cyprus are exclusively responsible for their internal affairs and do not exercise state powers, while for its part the state has recognized that they possess wide-reaching jurisdictions and has no right to intervene in their internal affairs". When issues of common interest arise, the competent Authorities discuss them with the five major religious communities from a position of equality. The latter make recommendations, but the State is exclusively responsible for making the final decision. The other religious communities (apart from the major five) are full beneficiaries of the right to religious freedom under the provisions of article 18 of the CC but do not enjoy the special privileges accorded to the five major religions.

*2.2. Greece*

In Greece, relations between Church and State are defined in article 3 of the 1975 Constitution, while article 105 affirms the ancient, privileged regime of Mount Athos. Concurrently, article 13 enshrines the freedom of religious conscience. The aforementioned article 3 establishes the Eastern Orthodox Church as the "prevailing religion" in Greece (Robbers 2019, pp. 171–94). The exact regulatory scope of this term has been the source of much theoretical discussion (Konidaris 2020, pp. 145–57) According to the first, minority view, which reflects the assumptions of the 1952 Constitution, the term is taken to mean the official State religion. However, the current prevailing view is that the term merely signifies the religion observed by the majority of the Greek people and is of limited regulatory significance. Finally, according to a third opinion, adopted by the Council of State in cases it has heard on the issue of lessons on religion in secondary education, and which in our view is the most accurate, the term "prevailing religion", apart from expressing a statistic,

functions as an interpretive tool for other constitutional provisions.[6] Article 3 establishes two systems on the foundation set by the enshrinement of the prevailing religion, which contextualizes the right of religious freedom: homotaxy between the Hellenic Republic and the Ecumenical Patriarchate on the one hand (Venizelos 2000, p. 112) and a mild system of State supremacy in the relations between the Republic and the Autocephalous Church of Greece. According to what is known traditionally as the system of the "legally ruled state" (νόμῳ κρατούσα Πολιτεία), the State intervenes and regulates the administrative matters of the Church. Additionally, the Charter of the Church of Greece is a formal law that must be voted by the Plenary Session of Parliament, no less. More precisely, the Charter states that, as far as their legal relations are concerned, both the Church of Greece and its formal structures (metropolises, parishes) constitute legal entities governed by public law (Konidaris 2020, pp. 397–450). The other religions are protected under the provisions of article 13 of the Constitution of Greece, while Law No. 4301/2014[7] regulates religious communities.

*2.3. The Safeguarding of Religious Freedom as the Common Denominator in the Legal Orders under Examination*

At the national level, religious freedom is enshrined in constitutional provisions in all two of the states under examination, as well as in the ECHR and the Charter of Fundamental Rights of the European Union. In addition to religious conscience (forum internum), which is inviolable regardless, the freedom to manifest religious beliefs (forum externum) and the freedom of worship are also established, though these are subject to restrictions. Both individuals and religious communities themselves are beneficiaries of this right (ECHR 2022). A sub-system of religious freedom is the establishment of the autonomy of religious communities (Robbers 2001, pp. 1–35; Doe 2011, p. 114). As a result of their autonomy, religious communities are (and must be) administered in accordance with their internal regulations, while the individual rights of their members may be limited by these regulations if and to the extent that inclusion and continued participation of individuals in such a religious community is founded on the free will of the faithful. For example, the prohibition on the ordainment of women as presbyters and bishops in the Orthodox Church[8] (Konidaris 2020, pp. 174–81) or the complete exclusion of women from the ranks of the clergy in the Catholic Church does not constitute a breach of the principle of equality between the sexes and is not unfair discrimination. Furthermore, the slaughter of animals as part of a religious ceremony is exempt from the general regulatory framework on slaughtering animals.[9]

Of course, the extent to which religious freedom is enshrined depends on the current network of relations between the State and Religions. Thus, the system of State supremacy established by the Greek Constitution provides a contextual framework for the religious freedom enshrined in article 13 (Vlachopoulos 2022, pp. 254–63). Article 105 serves a corresponding purpose but only pertains to Mount Athos (Konidaris 2020, pp. 365–74). The system of homotaxy exhibits lesser influence. It is based on coexistence and tolerance and places significant emphasis on the self-government of religious communities. Furthermore, the nature of religious groups, which the Constitution recognizes as legal entities governed by private law, means their detachment from state authority on the one hand and their ability to develop for-profit activities on the other. The Orthodox Church in particular has business activities and exerts a significant degree of financial power and social influence. Naturally, this leads to exponentially increased independence and a stronger bargaining position. Thus, the otherwise equal relations between Church and State, albeit with a slight state supremacy in decisions, are interpreted and implemented in light of the above actual situation. Compared to the Greek paradigm, there is a clear reversal, which strengthens the position of the Church. Before Cyprus achieved independence, the Church led the Nation and even possessed political competencies as part of the Ottoman Imperial administrative system, a status quo that was maintained into the period of English rule.

As shall be discussed in the next section, the extent to which the system of relations between the State and Religions protects religious freedom is reflected in the harmonization of the right with the protection of public health.

### 3. Religious Communities and the Pandemic: From the Imposition of Restrictions to Worship to Potential Conflict with the State in Their Self-Regulation

*3.1. The Imposition of Restrictions on Worship and the Reaction of Religious Communities*

For reasons of expediency, the time period of restrictive measures imposed and the reactions of religious communities must be common for all three states under examination and will extend to January 2021, despite the fact that they did not implement or abolish restrictive measures on the same dates. Particular focus will be given to the major events which provoked social unrest and led to the formulation of case law. Furthermore, as far as Cyprus and Greece are concerned, the focus shall be on the Orthodox Church, which is the largest of the five privileged religions in Cyprus and is the prevailing religion of Greece. Indeed, it is because of these that we have information available.

3.1.1. Cyprus

In Cyprus (Kastanas 2021, pp. 120–26), the first phase of tangible restrictive measures against the coronavirus was introduced with the address[10] of the President of the Republic on 13 March 2020. Two days later, the Council of Ministers adopted the measure restricting social gatherings in larger interior spaces, including places of worship—and by extension the parish churches which constitute the subject of the present paper—to a maximum of seventy-five (75) people. It must be noted that in the announcements which followed, the responsibility for announcing and, to an extent (as demonstrated ex post), specializing protective measures during congregations was delegated to the Church itself. This choice of the Executive branch must not be considered a mere gesture of respect and courtesy but must be viewed instead as a result of the system of relations between the State and Church and the increased autonomy of the Church, which manifested itself further in the subsequent phases of protective measures.

The press release of the Holy Archbishopric of Cyprus which followed (Archdiocese of Cyprus 2020) contains (in addition to the obvious, i.e., a call to prayer) a number of items worth further mention. To begin with, the official church not only fully harmonized with the restrictive measures but also threw its full support behind the executive branch, the decisions of which it endorsed in full. Furthermore, it called on the faithful to abstain from worship, "because the benefit to be gained after this trial has passed shall be far greater" (Archdiocese of Cyprus 2020). A final, but not insignificant aspect was the acceptance of "digital worship"[11] and suggestions that the faithful content themselves with it during this period of refrainment from in-person worship.

In this first phase, refraining from in-person worship was a recommendation and not mandatory and led the Church[12] to adopt another, informal measure: it allowed up to ten (10) of the faithful to enter churches and observe services and burdened church committee members with enforcing this restriction by monitoring the number of faithful. This initial period of milder restrictive measures was short-lived, however, as the spread of the coronavirus forced the executive branch to institute far more wide-reaching and severe measures, which also affected places of worship. According to the Decree of the Minister of Health, issued pursuant to the law on quarantines and with the objective of protecting public health to limit the disease caused by the COVID-19 Coronavirus and avoid a potential collapse of the public health service due to an uncontrolled spread of the virus, citizens were prohibited from entering areas of religious worship such as churches, mosques and other prayer spaces (Kastanas 2021, p. 119). Subsequently, this prohibition was extended to the 3 May 2020.

In this period, which included Holy Week and Easter Sunday, church services were held, but "the doors were shut"[13] to the faithful, and only those necessary to perform the services were permitted in (priests, chanters, altar servers and some church committee mem-

bers), while church services were broadcast over television and radio and live-streamed on the internet wherever the necessary infrastructure was in place. Nevertheless, churches never stopped functioning throughout this period, even if only in this manner. The 4 May 2020 was a major landmark for public worship in Cyprus: it was the day on which the prohibition of the faithful to enter places of worship was lifted, and the terms and preconditions in accordance with public health protocols regarding protection from the coronavirus in large, enclosed spaces were determined. The object of the current discussion, the Church of Cyprus, allocated the implementation of these protocols to the church committees of the individual parish churches. In fact, at the highest echelons, the Head of the Church of Cyprus communicated with the President of the Republic and personally guaranteed faithful observation of the protective measures. A notable example of self-organization was the adoption (by the Archbishopric, at least) of more stringent social distancing measures within churches and church courtyards in relation to the prevention measures and directives regarding the novel coronavirus issued by the competent Ministry (Church of Cyprus n.d.). More particularly, while the State connected the number of the faithful with church space in square meters, the Archbishopric applied a horizontal standard criterion: Church services would be held with open doors, but church committee members would be burdened with ensuring that the number of faithful within any given church, regardless of size, did not exceed ten (10) at any time. Given that parish churches were generally quite spacious, this was a stringent measure, albeit one which served to demonstrate the decisiveness of the Church administration. Similar measures were adopted for the faithful who wished to receive the Holy Communion. They were to appear at a strictly set time at the church, and the church committee members were to ensure that the total number of faithful within the church to receive Communion did not exceed ten (10), all while maintaining the social distances mandated by law. Once a member of the congregation had received Communion, they were to exit the church and the next person in line was to enter, etc. Naturally, this meant a significant increase in the duties of the church committee members, who were burdened with the unenviable task of implementing the measures and monitoring their appropriate observation by the faithful. This confirms the active participation, even during a period of pandemic, of laypeople in the function of the Church organization, which was always increased (Konidaris and Emilianides 2016, pp. 267–80) in comparison to the other Orthodox Churches. These measures were supplemented[14] by the mandatory use of protective face masks, which up until that point had been optional with the recommendation that members of the congregation who belonged to vulnerable groups wear them.

The legality of the restrictive measures was not contested before the Courts of Cyprus. However, case law was produced regarding accusations of violations of the provisions of the law on quarantine regarding gatherings of more than two individuals, as well as on incitement to the committing of an offence. In particular, the Metropolitan of Morphou was prosecuted on the accusation of inciting the faithful to participate in a public meeting, in particular, in the blessing of the waters at the Karkotis river on 6/1/2021. As noted in the decision (News Portal Aggeliaforos 2023) "[...] From the testimony before me, the criminal behaviour for which the Accused is present here today, i.e., the invitation to a public meeting or congregation of more than 2 individuals, is not substantiated", as "there was an invitation and a calling, but to a permitted act of religious worship or, at the very least, an extension of the daily congregation in accordance with Regulation 2(1)(θ) of Regulatory Administrative Act 615/21, on the day the invitation was made, 6/1/21." To summarize, the Judge notes that: "...[...] the blessing of the waters of the Karkotis river was not a public meeting but an activity which the legislation itself, 'in its wisdom' as it were, permitted exceptionally on 6/1/21, the day of the Theophany. Legislators mandated this exception in a clear recognition of the significance of this particular religious ceremony and as a means of relaxing the myriad restrictions for reasons of public health which had been imposed on the faithful of the island".

### 3.1.2. Greece

In the case of Greece, a distinction must be made between the period before the first lockdown, the lockdown period itself, the period of limited worship and the second lockdown.

An Act of Legislative Content issued in February 2020 provided permission to implement restrictive measures to protect public health, concurrently instituting the extraordinary legal remedy of objections before the Ordinary Administrative Courts. Meanwhile, the prime minister published a tweet decrying the restrictive measures adopted by decision of the Standing Holy Synod of March 2020 as inadequate. Subsequently, a Joint Ministerial Decision established provisional restrictions on the performance of all forms of religious service in all spaces of religious worship for precautionary public health reasons, until 30 March (Androutsopoulos 2020, p. 47). The faithful were strictly permitted to enter churches for personal prayer and to perform funerals, with the presence of a priest and only close family members, while liturgies were held with the sole purpose of being broadcast via television, radio or the internet. The restrictive measures were extended though an amendment permitted the participation of necessary staff in the services of Holy Week and Easter.[15]

At the end of the universal lockdown in May 2020, places of worship were allowed to function again, albeit with restrictions on the number of the faithful per square meter and the requirement to comply with all the protective measures implemented by the National Committee for Public Health Protection. These measures were extended throughout the summer, while in view of the Feast of the Dormition of the Virgin (15 August), which is traditionally commemorated with litanies, all religious services of all faiths and creeds performed via processions outside of spaces of religious worship were suspended (Androutsopoulos 2020, pp. 51–52).

The increase in coronavirus cases in November 2020 led to the reinstatement of the restrictive measures implemented in the second period. This time, however, after conferring with the Church of Greece, the Government permitted the participation of the faithful in the celebrations for Christmas and New Year's Eve 2021, though with strict observance of the public health measures (Press Release of the Church of Greece 2020). This initially covered the celebrations for the Theophany as well, albeit without the performance of a crucession or the blessing of the waters as mandated by the typikon of the Orthodox Church. The government cited worsening epidemiological conditions for its ultimate decision not to allow the participation of the faithful in the services. The Holy Synod reacted to this and decided to open the churches, implementing the Christmas and New Year's Eve measures in clear breach of the law while concurrently bringing their case before the Council of State.[16] The faithful, as was to be expected, followed the decision of the Church and not the State mandates, while the police, clearly acting on the orders of the competent Minister, did nothing to prevent services from being held.

The above restrictive measures on worship were brought before the courts of Greece. The regulatory decisions and the ad hoc imposing of measures were challenged, in accordance with the applicable procedural provisions, before the Council of State and the ordinary administrative courts. In decision no. 49/2020,[17] the Suspensions Committee of the Council of State ruled that the temporary nature of the measures together with the reasonable duration of their enforcement in conjunction with the inability for other effective measures to be implemented immediately for the protection of public health all but precluded their suspension, due to pressing reasons of public interest. Decision no. 60/2020[18] issued by the same body noted that the Executive branch was required, in accordance with the principle of proportionality, to justify the necessity of the measures in order to deal with the pandemic, in light of the fact that these severely limited the ability of citizens to exercise their individual rights, such as those to religious freedom and, in particular, freedom to worship. Additionally, it ruled that such measures were subject to judicial review. Put simply, the Council of State emphasized the principle of proportionality and the requirement for justification for the implementation and extension of the measures. However, due to the procedural system in place, the Council of State did not examine the

merits of the case at the level of final judicial protection,[19] instead preferring a rather more formalistic approach. More particularly, the Court ruled that the parties did not possess special legal interest for the continuation of the proceedings, this being moral damage caused by the contested JMDs, in particular, the core of the right to freedom of religion as expressed by the freedom to worship (Kapsali 2020, pp. 46–54). According to the court, the faithful could claim compensation for the increased mental and emotional trials they were presented with due to their isolation from the religious community via a relevant lawsuit, without the need to bring an application for annulment before it.[20]

The decisions of the administrative courts of the first instance[21] toed the same line, for the most part. In particular, regarding the prohibition of litanies on 15 August, the courts ruled that, given the provisional nature of the measure, there was no issue of State intervention in the internal affairs (interna corporis) of the Church. This was because the contested joint ministerial decision did not abolish litanies permanently but instead prohibited them for a notably brief period of time (throughout August 2020), as a relevant restriction to the freedom of religious worship in order to protect the supreme good of the public health of all individuals living in Greece.

Apart from the court decisions which ruled on the legality of the restrictive measures imposed on worship, it is also worth noting the decisions of the single-member and three-member Misdemeanour's Court of Corfu (Press Release of the Metropolis of Corfu 2023), which declared, both at the first instance and on appeal, the Metropolitan of Corfu innocent of the accusation that he had told the faithful in his sermon that they could visit church to pray or receive communion with a permit for physical exercise and that he had performed the holy liturgy and procession with the relic of St Spyridon within the church on Palm Sunday in April in the presence of more participants than permitted by the government decisions. The metropolitan was declared innocent on the basis of article 33 of the new Criminal code, which allows the courts to take into consideration the issue of conflict of duties natural persons encounter in fulfilling their duties when doing so results in a breach of the law.[22]

The other religions In Greece also conformed to the imposed measures without having any say in the nature of the restrictions, which belonged to the exclusive competence of the Greek State. The current regulatory framework does not make provisions for suggestions by religious communities (including that of the Prevailing Religion), while waiting until the religious communities issue decisions or even taking these decisions into account is down to State goodwill. Management of the restrictions to worship from a communication standpoint, as well as the political fallout of any such decision among religious communities, is another issue entirely.

### 3.2. Justification of the Measures on the Basis of the Internal Regulations of the Religious Communities

Regarding the internal regulations of the Orthodox and Catholic Church, i.e., the Canon Law of each denomination, restrictive measures were based on the concept of church economia. According to Orthodox Canon Law, but also to Roman Catholic Law, the term economia (oikonomia) denotes a timely and logically defensible deviation from a canonically established rule for the sake of bringing salvation either within or outside the Church. But this deviation does not extend to the point where it could violate the dogmatical boundaries of the rule in question. Also, economia should be decided upon only by the canonically instituted authority of the Church. It should be kept in mind, however, that economia is an extraordinary ecclesiastical measure, the nature of which is timely and its duration temporary. Its intent is solely Christian, an expression of the love that guides the Church and of the virtues that issue from it, sympathy, leniency and understanding of human weakness. But no dogmatical boundaries should be moved or removed when economia is applied (Feraru 2018, pp. 118–19).

It is worth noting that no decision—by either the Church of Cyprus or that of Greece—to implement restrictive measures on worship mentions the principle of church economia explicitly (Giangou 2021).

However, there can be no doubt that this principle was applied in practice. (Giangou 2021). The extraordinary public health situation necessitated the suspension of worship or, at the very least, modifications to the liturgical typikon. The hallmarks of the implementation of economia were reflected in this temporary situation. The governing Church, through the wording of the decisions that modified its rite and its tacit acceptance of state restrictions on worship, was applying economia even if not stating as much expressis verbis. This may be explained by the nature of the principle itself and how the Orthodox Church has applied it throughout history (Papathomas 2016, pp. 126–30). Despite being an ancient principle of Canon Law, its prerequisites have never been defined in a clear, unambiguous manner. Indeed, it is exactly this ambiguity which prevents it from being identified with the corresponding principle of the Codex Iuri Canonici, and in any case its legal classification is not the crucial issue. Put simply, it lacks scholasticism. Orthodox Canon Law has not been codified and is thus applied with a certain liberality, in accordance with the homily of Athanasius the Great.[23] As is to be expected, the means of applying the principle are influenced by the system governing the relations between Church and State. Concurrently, however, the very acceptance itself of a system of State supremacy, in which the State intervenes in matters of Church administration, could be viewed as a form of church economia, or at the very least as a form of sustained compromise by the Church. As regards the pandemic, this compromise was maximized in both extent and intensity. However, in light of the extraordinary public health conditions and the similar measures enacted by the State and many other States regarding other aspects of public life, the extent of the compromise was defined by the current system of Church–State relations.

As for the initiative to implement restrictive measures to worship, the religious communities of Cyprus possessed a higher degree of freedom to self-regulate than those in Greece. This may be attributed both to the system of homotaxy and to the fact that, at least concerning the Church of Cyprus, relations between the President of the Republic and the Archbishop of Cyprus were quite cordial.

Another noteworthy aspect discussed and forced to change during the pandemic was the traditional practice of the faithful receiving Holy Communion with a common spoon (Kessareas 2023). The Church administrations of both Greece and Cyprus refused to consider changing this tradition in the period in which participation in worship at church was still permitted (Kessareas 2023).

## 4. Towards a Methodology of Harmonization between Religious Freedom and Extraordinary Public Health Interest

The timeline of adoption and implementation of restrictive measures on worship during the pandemic reveals a methodology of harmonization between religious freedom [in both its individual and its collective aspects as well as in light of the autonomy of religious communities] and public health interest. The next section entails an examination of whether the system of relations between Church and State impacted how this methodology developed.

To start with, the harmonization framework itself was an extraordinary, emergency situation. As such, the exercise of individual rights and their corresponding state competencies could not be viewed within the bounds of normality. But while this was viewed as true during the initial months of the pandemic, it could hardly be expected to last into the future and until the restrictions were lifted. The exception became the norm, and thus any restrictive measures, and restrictions in general, required strict, detailed and exhaustive justification and oversight to ensure the guarantees of the rule of law. Especially regarding religious freedom during the pandemic, the relevant right could be viewed in a completely new light, by definition restricted. The forum externum, far from being restricted, was instead delineated and given new meaning by the need to protect public health. In other

words, it was public health that dictated what was and was not permissible. In light of this, proportionality assumed increased importance, and, indeed, proportionality tests were required to be more stringent the longer the restrictive measures were in place. Justification for state measures was required to be comprehensive, clear, specific and sufficient in order to permit proportionality tests.

This requirement for stringent proportionality tests is not stated emphatically in the—in any event limited—Greek and Cypriot case law. By contrast, both the French Conseil d'État and the German Bundesverfassungsgericht focused on the proportionality aspect and unflinchingly emphasized the unconstitutionality of the restrictive measures. However, it was none other than the U.S. Supreme Court that established the relevant framework and demonstrated just how important it was to test the proportionality of restrictive measures. The resultant rules are also applicable to the legal orders under examination despite being imported, so to speak, as they reflect general principles of the restriction of individual rights. This new right is ephemeral and functions according to two axes. As discussed previously, the new right applies as long as restrictive measures are in place. Concurrently, as these measures subside, aspects of the previous right to religious freedom are strengthened correspondingly, in such a way as to reinstate the right to religious freedom as it applies under normal circumstances once the restrictive measures are abolished in full.

## 5. Conclusions

The pandemic gave rise to a new means of harmonization between religious liberty and the protection of public health. Within this context, religious liberty and, in particular, the freedom of worship may be subjected to legitimate restrictions with respect to the principle of proportionality. The autonomy of religious communities is redefined within this temporary harmonization, as they are called on to adapt to the developing situation. It could be concluded that this osmosis gave rise to a new perception on religious autonomy, impacted and defined by societal and public health imperatives. However, self-regulation is dependent on the existing system of relations between the State and Religions. The two cases discussed herein demonstrate that greater autonomy is observable the further away we move from unitary systems of state supremacy and closer to systems of separation. However, beyond the institutional framework, relations between the heads of State and Religion had an observable de facto impact on the success of the harmonization. This balance gives rise to general principles, the implementation of which must always be timely and connected directly to emergency situations, the management of which also serves the field of religious freedom.

Besides the different system of the relations between the state and denominations, the comparison of the two cases, despite their multi-level resemblances, with the most basic being the common Orthodox tradition and the similar enshrinement of religious freedom, does not permit of the extraction/inference of a common rule of the restriction of religious freedom under emergency circumstances. Each case retains its own proper character. What is only certain is the subjection of the case of the restriction of freedom to derogation status, whether it is seen from a philosophical concept or as a nexus of posited legal rules which regulate legal relations in crisis periods. This is finally also the beneficial outcome from the study of the two cases. The subjection or non–subjection to such status, besides a theoretical scholarly discussion or an analysis within a religious community, is the duty of the Judge with the principle of proportionality as an instrument. The peculiarity of the COVID-19 pandemic lies in the fact that the public health need was so surprising, compelling and severe that the restriction reached the very core of the right. In Greece where related cases were brought before the Court, but also in Cyprus to a limited extent, the cases were judged in formal *res judicata* effect at a subsequent time point with the result that public health data had changed by then. Thus, the decisions could not *de facto* have been enforced. In any event, the experience from the harmonization of religious freedom and public health constitutes a significant precedent for the treatment of a future pandemic.

**Funding:** This research received no external funding.

**Institutional Review Board Statement:** Not applicable.

**Informed Consent Statement:** Not applicable.

**Data Availability Statement:** No new data were created or analyzed in this study. Data sharing is not applicable to this article.

**Conflicts of Interest:** The author declares no conflict of interest.

## Notes

1.  The issue of the harmonization of religious freedom with the protection of public health is not novel. On the contrary, it is old and is based on the premise that freedom of belief may be absolute, but freedom of action is not. For example, based on this premise, the US Supreme Court, in its famous ruling in Jacobson vs. Massachusetts (1905), introduced the principle of mandatory vaccination—a case law affirmed some years later in the Court's 1922 ruling in Zucht vs. King.

2.  The Eastern Orthodox Church clearly lacks the uniform, hierarchical structure of the Roman Catholic Church, instead being governed by a system of Patriarchates, Autocephalous and Autonomous Churches distinguished by their comprehensive (or limited, in the latter case) right to self-govern.

3.  They are articles 23§§9-10,110,11 of the 1960 Constitution.

4.  In correspondence with the provisions of the Constitution of Greece. See next section.

5.  This was hardly an innovation of the CC, which was based on the system in place during the British colonial period, itself based on the system introduced by the Hatt-ı Hümayun (Ottoman Reform Edict of 1856). Clearly, Islam was no longer accorded a privileged position vis a vis the other religions.

6.  In the case regarding lessons of religion, their denominational nature was established on the basis of a combined reading of articles 3 and 16(2) of the Constitution.

7.  L. 4301/2014, Nomos Database.

8.  By contrast, the ordainment of female deacons is permitted according to Canon Law and is observed by some Churches (Patriarchate of Alexandria).

9.  ECJ Case C-336/19. Available online: https://curia.europa.eu/juris/document/document.jsf?docid=235717&doclang=EN (accessed on 7 July 2023).

10. Statement by the President of the Republic, Mr. Nicos Anastasiades, following the extraordinary meeting of the Council of Ministers' ('Δήλωση του Προέδρου της Δημοκρατίας κ. Νίκου Αναστασιάδη κατά τη συνέντευξη Τύπου, στο Προεδρικό Μέγαρο, μετά την έκτακτη συνεδρία του Υπουργικού Συμβουλίου). Available online: https://www.pio.gov.cy/%CE%B1%CE%BD%CE%B1%CE%BA%CE%BF%CE%B9%CE%BD%CF%89%CE%B8%CE%AD%CE%BD%CF%84%CE%B1-%CE%AC%CF%81%CE%B8%CF%81%CE%BF.html?id=12670#flat (accessed on 4 August 2020).

11. This neologism is an attempt to give a name to the participation of the faithful in liturgies through live broadcast of the church services via television or radio or streaming on the internet. Transmissions such as these are as old as the means on which they are transmitted and have been available ever since. However, in the pre-COVID period, the rule was that "listening to Church Services and the Divine Liturgy via broadcast cannot in any way substitute for the physical presence and participation of the faithful in Parish life". Available online: https://ecclesiaradio.gr/ (accessed on 4 August 2020).

12. At least, to churches within the administrative purview of the Holy Archbishopric of Cyprus, according to first-hand observations.

13. John 20,19.

14. Beginning on the 1st of August 2020, in accordance with the relevant Decree of the Ministry of Health due to the rise in documented coronavirus cases in Cyprus.

15. It must be noted that the Government took into consideration the pastoral concerns of the Church, communicated via the Synod.

16. Available online: https://www.kathimerini.gr/politics/561216136/ieri-antarsia-gia-theofania-sto-ste-prosfeygei-i-diarkis-ieria-synodos/ (accessed on 5 July 2023).

17. See the Council of State judgment no. 49/2020, Nomos Database.

18. See the Council of State judgment no. 60/2020, Nomos Database.

19. See the Council of State judgment no. 1294-96/2020, Nomos Database.

20. The rulings of the Fourth division of the Council of State issued after those of the Suspensions Committee had a three-to-two majority, with the minority opinion holding that there was no procedural inadmissibility and that the Court was obligated to examine the merits of the case.

21. See the Administrative Court of the First Instance judgment no. 342/2020, Nomos Database.

22. This was the case with the Metropolitan of Corfu, who felt that his prompting of the faithful to receive communion in the midst of the pandemic stemmed from his deep faith in God and his position as a hierarch of the Church of Greece as well as his

commitment since youth to the Church, and his performance of the litany with the relic of St Spyridon within the holy church on Palm Sunday, 12 April 2020 with the minimum required number of priests and assistants, despite this conflicting with the express letter of the law as stipulated in the JMD imposed by the State, was what his conscience dictated.

23 "The letter kills, but the Spirit gives life. Indeed, if we were to read only the letter of much of the holy Texts, we would find ourselves submerged in blasphemy." Athanasius the Great. "Τεμάχια εκ των εις το κατά Ματθαίον", PG 27,1384B.

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
