# Peer review of "The Harmonization between Religious Freedom and the Protection of Public Health: Betwixt Self-Regulation and Law"

_religions, doi:10.3390/rel15020208_

Round 1

Reviewer 1 Report

Comments and Suggestions for Authors

The paper begins with a succinct systematization of RF legal framework in each of the countries but without much dialogue between them. Initially, it would be also necessary to go through a state of the art, placing this article in the field of the many works that, in recent years, have been done on the challenges to RF during the pandemic.

It would be necessary to justify the reason for choosing these 3 case studies and explain the concrete elements that make them (in)comparable. If, on the one hand, the comparison between two countries with an orthodox majority and historical similarities seems more or less obvious (something that inversely tends to make it difficult to maximize the differences between them); on the other hand, the comparison with France, an exceptional country for the originality of laïcité and for the complex relationship with sects, seems more difficult to justify.

If, on the one hand, I understand the need to systematize and summarize the analysis of the selected countries; on the other hand, this hyper systematization, analyzing each country individually, without a clear articulation between them, makes it seem that the article is divided into three parts and that it is not a complete object. This would be a good methodology for a thesis and/or book in which selected countries needed to be thoroughly justified.

The article has a strong descriptive and little analytical dimension. I suggest reducing the former and investing in the latter in order to reinforce the article.

The article lacks theorization and debate with some important theorists such as Peter Berger (Many Altars of Modernity), Charles Taylor (Secular Age), Jorg Stolz (Religious-Secular Competition), Gert Pickel (Contextual Secularization), Alfred Stepan (Multiple Secularisms), Rajeev Bhargava (Principled Distance) and, above all, Monika Wohlrab-Sahr and Marian Burchardt (Multiple Secularities). Without the debate with at least two or three of these ideas and the consequent problematization of the object of study, the work does not have the potential it could reach.

I would suggest that, overall, the work focuses more explicitly on the forms, impacts, and successes of harmonization, as this is one of the most original contributions of the paper.

Author Response

With the new version of my article, I limited myself only to Greece and Cyprus, which are Orthodox countries, with a common culture. Each State, however, presents a different system of State-Church relations. I only focused on the legal side of the issue and not the sociological side of the issue. I tried to delve deeper into the harmonization of religious freedom with the protection of public health from the perspective of different systems of State-Religion relations.

Kind regards,

Reviewer 2 Report

Comments and Suggestions for Authors

As the lessons of the State's response to the health crisis are still being learned, the challenge posed by the COVID-19 pandemic to the protection of freedom of religion or belief will be debated in legal scholarship for a long time to come. The subject the writer is dealing with is therefore of great importance. However, the design of the research seems to me to be poorly conceived and, in any case, without sufficient foundation. 

The author has chosen three countries whose experiences are to be studied: Greece, Cyprus, and France. These are two Orthodox-majority countries and one generally considered to 'represent' secularism. What is meant to emerge from juxtaposing these three? If the author's intention was to compare how different Orthodox-majority countries dealt with the pandemic, France is superfluous. If the aim was to compare state responses in the Orthodox world and in the home of laïcité, Greece or Cyprus are superfluous. And if the author is interested in drawing conclusions that are potentially valid for all of Europe, it is hard to understand why Catholic and Protestant countries were excluded from the study. The author's only statement on the article's objective seems to point to the last option, the most ambitious one ("to identify by induction the general principles of harmonisation between religious freedom and the protection of public health"). This is also confirmed by the last two sections of the article.

The conclusions contain findings that are not supported by earlier considerations. What does it mean that "[t]he autonomy of religious communities is redefined within this temporary harmonisation, as they are called on to adapt to the developing situation"? Is this alleged redefinition in accordance with international law? Does it apply to other states? Is it permanent, going beyond this temporary harmonisation? The author also claims that the "relations between the heads of State and Religion had an observable de facto impact on the success of the harmonisation", but in the previous sections of the manuscript there is no mention of the French President.

Judging by the list of references, the author is Greek or Cypriot. Perhaps it would be better to focus on Greece and Cyprus, with an emphasis on the role of the Orthodox Church, rather than making a superficial analysis of French law and attempting to answer general questions on the basis of this scant material? This topic is still underrepresented in the legal literature and I would strongly encourage the author to fill this gap. On the other hand, if the author insists that the article should develop general principles for possible use in other jurisdictions, then much more sources should be consulted, in particular the special issue of 'Laws' (https://doi.org/10.3390/books978-3-0365-2280-7), as well as the results of international research programmes such as the Oxford Compendium of National Legal Responses to Covid-19. 

The concept of homotaxis should be explained to the reader in a footnote. 

Comments on the Quality of English Language

This manuscript requires thorough proofreading by a native English speaker before it can be resubmitted. 

Author Response

(The authors gave the same response as above.)

Round 2

Reviewer 1 Report

Comments and Suggestions for Authors

The author maintains references to France in the abstract and in section II. Given the reconfiguration of the article, this should be deleted.

For the rest, an effort has been made to strengthen the analysis of the part relating to the harmonization of the RF. This reinforces its analytical dimension, but I don't know if it's enough to be more than an essentially descriptive article.

I regret that none of the bibliographical references I suggested were included.

Author Response

Thank you for your comments. I tried to make use of them.

Reviewer 2 Report

Comments and Suggestions for Authors

Honestly, it was a rather disappointing paper from the very beginning, but I nevertheless gave it a chance in the hope that the author would be able to improve it. I regret to say that the revision submitted for the second round of peer review does not live up to these optimistic expectations. The corrections have been made in a hasty and chaotic manner, with no regard for the coherence of the final result. The author declares that he or she has decided to remove France from his or her research. And yet France keeps appearing: in the abstract, in the title of Section II, and even in the list of references, where sources on France have been kept, even if they are not used in the body of the article. The conclusions still refer to "the three cases discussed herein". Even more surprisingly, the conclusions have not been modified at all, even though the focus of the paper has changed. The author has not even bothered to expand the bibliography to include the items I suggested. I therefore see no reason to recommend this piece for publication. 

Comments on the Quality of English Language

This manuscript could definitely benefit from some deep editing. For example, what does it mean that any restrictions "should pass through the system of Church-State relations"? 

Author Response

(The authors gave the same response as above.)
